# Soft Checksums to Flag Untrustworthy Machine Learning Surrogate Predictions and Application to Atomic Physics Simulations

## Abstract

Trained neural networks (NN) are attractive as surrogate models to replace costly calculations in physical simulations, but are often unknowingly applied to states not adequately represented in the training dataset. We present the novel technique of soft checksums for scientific machine learning, a general-purpose method to differentiate between trustworthy predictions with small errors on in-distribution (ID) data points, and untrustworthy predictions with large errors on out-of-distribution (OOD) data points. By adding a check node to the existing output layer, we train the model to learn the chosen checksum function encoded within the NN predictions and show that violations of this function correlate with high prediction errors. As the checksum function depends only on the NN predictions, we can calculate the checksum error for any prediction with a single forward pass, incurring negligible time and memory costs. Additionally, we find that incorporating the checksum function into the loss function and exposing the NN to OOD data points during the training process improves separation between ID and OOD predictions. By applying soft checksums to a physically complex and high-dimensional non-local thermodynamic equilibrium atomic physics dataset, we show that a well-chosen threshold checksum error can effectively separate ID and OOD predictions.

## 1 Introduction

An ability to detect errors would increase trust in machine learning (ML) surrogate models to make scientific predictions. In critical applications, unnoticed errors can have severe consequences, leading to inaccurate conclusions and poor engineering design choices. These errors are often caused by applying a surrogate model on data points where it is not a valid approximation of the true function. For a given surrogate model, there exists a domain of validity where you can reliably characterize how the network behaves due to the use of a validation dataset during training. We aim to develop a metric that excludes predictions when we are not confident that the surrogate model is reliable.

Uncertainty quantification attempts to distinguish between the often unknown domain of validity (or validation domain) and the domain of intended use when discussing computational physics surrogate models (Oberkampf et al., 2004; Riedmaier et al., 2021; Roy & Oberkampf, 2010). While the domain of validity has boundaries based on the physical experiments or simulations conducted to populate the validation dataset, the domain of intended use refers to all physical states the user may apply the surrogate model to. Ideally there is total overlap, but in reality it is difficult for a user to define the boundaries, and often portions of the domain of intended use are outside of the domain of validity. While physics-based models may have some level of confidence in regions outside of the domain of validity due to a deep understanding of the system, the user can only rely on an estimation of the boundaries and extrapolative capability.

This same idea exists in machine learning, which often views data points as sampled from an unknown distribution and correspondingly refers to the domain of validity as the set of in-distribution (ID) data points, and all other data points as out-of-distribution (OOD). While still difficult to detect in practice, the difference between ID and OOD data in classification problems with discrete outputs can be simpler to visualize as the OOD data is often from entirely different datasets or classes. How-

ever, when a classification model attempts to identify a blurry image, or a regression model maps inputs to a continuous, potentially high dimensional output space, there is a clearer analogy to computational physics and the similarly unknown boundaries between ID and OOD data. Specifically for regression problems, all predictions will have non-zero error, eliminating the binary evaluation of correct or incorrect. Instead, we want to differentiate between ID predictions with acceptably low error and OOD predictions with unacceptably high error, a challenging task due to the complex, application dependent boundary.

We can consider ML surrogates as both uncertain physical surrogates (with a domain of validity separate from their domain of use) and as statistical ML models (trained on ID data, but expected to encounter OOD data). An ML surrogate offers the advantage of providing faster results with less computational cost (Almeldein & Van Dam, 2023; Carranza-Abaid et al., 2020; Ganti et al., 2020; Kluth et al., 2020), but this comes with the risk of unnoticed errors propagating and rendering the simulation useless. In particular, it would be helpful if we could detect when the model is being asked to predict outside the domain of validity (i.e. on OOD data). We could use this information to decide whether to trust our final results, or to revert back to detailed and expensive physics sub-simulations to preserve the reliability of the overall simulation.

Our goal is to provide information to help the user by raising a nominal red flag if an ML surrogate model is likely predicting on OOD data and should not be trusted in scientific regression applications. A naive approach might assume that a trained surrogate is able to predict with sufficient accuracy on any data point within a hypercube bounding the training dataset. However, this is likely not valid for physical problems. Collecting data from trusted simulations or experiments likely does not produce an evenly sampled distribution to populate the training dataset. Especially in high dimensions, this could result in gaps where data points may be physically possible, but are not well represented.

The main contribution of this work is a novel checksum based method for indicating untrustworthy predictions due to OOD data. Similar to checksums in message transmissions as described in Section 2.2, we add an additional output to the surrogate model and encode a checksum function. With this known relationship between the outputs, we can calculate a checksum error for each prediction. We can then differentiate between ID and OOD data as having low and high checksum errors respectively, and flag when the predictions should not be trusted. We refer to the encoded function as a soft checksum because while it is like a traditional checksum in that it can indicate potential errors, it differs in that it produces a continuous, rather than binary, signal.

We also propose a modified loss function, combining ideas from Physics Inspired Neural Networks (PINNs) (Raissi et al., 2019), fine-tuning (Liu et al., 2020), and Outlier Exposure techniques (Hendrycks et al., 2019). We use additional terms in the loss function to shape the checksum error surface and more consistently produce low values for ID predictions, and high values for OOD predictions. To achieve this goal, we implement a novel method for exposing a surrogate model to random OOD data during the training process, without biasing it towards a limited OOD region.

Importantly, using a soft checksum to flag untrustworthy predictions only requires a single model and forward pass, incurring negligible time and memory costs. This is a general method that makes no a priori assumptions about the data, and can be easily added to existing model architectures.

## 2 BACKGROUND AND RELATED WORK

### 2.1 OUT-OF-DISTRIBUTION DETECTION

There are several different methods that have been proposed for OOD detection and uncertainty estimates in regression problems. Bayesian methods are commonly used to approximate a posterior distribution with an uncertainty estimate, but can be limited by inaccurate priors and difficulty scaling to large datasets (Blundell et al., 2015; Fortuin et al., 2022; Neal, 1996; Wilson & Izmailov, 2022; Yang et al., 2019). Monte Carlo dropout has been shown to approximate Bayesian methods, but retains some of the same limitations with a slower training process (Gal & Ghahramani, 2016; Wang & Manning, 2013).

The most common alternatives to these methods are deep ensembles, which produce comparable, if not better, results than Bayesian methods by training multiple models and using the variance of

the predictions as a measure of uncertainty (Lakshminarayanan et al., 2017). However, this method involves training and evaluating multiple models, incurring larger computational costs. To avoid the need for multiple models and the complications of training a Bayesian neural network, recent work has focused on anchor based training which can produce an uncertainty with multiple evaluations of a single model (Thiagarajan et al., 2022; 2024).

Predating regression applications, there is considerable work developing OOD detection for classification problems (Yang et al., 2024). Most relevant for this paper are approaches which improve detection by including OOD data points in the training process. Both Outlier Exposure (Hendrycks et al., 2019) and an energy-based method (Liu et al., 2020) include explicitly OOD inputs in the training process by adding a term in the loss function which trains the model how to flag these data points.

## 2.2 CHECKSUMS

Checksums have been around for decades to verify data integrity, with Fletcher's Checksum being proposed in 1982 (Fletcher, 1982). The sender adds check bytes to the end of a transmission such that the calculated checksum should be zero. If the receiver does not calculate the same value, this indicates a transmission error and the message is resent.

However, the strict requirement of zero errors to satisfy a checksum is not always necessary. It is often sufficient to transmit images and videos with limited errors, or it may be more important to deliver the message and avoid the cost of retransmission rather than fixing corrupt bits. For those cases, it is more useful to estimate the fraction of corrupted bits with error estimating codes (Chen et al., 2010; Zhang & Kumar, 2017) or a soft checksum (Lee & Bahk, 2021), and allow the receiver to set a maximum threshold error, below which there is no need for retransmission.

In this paper, we bring these checksum concepts into machine learning for identifying prediction errors. As regression machine learning models will have non-zero error on predictions outside of the training dataset, a binary checksum method would flag all predictions. Instead, we borrow the term soft checksum and error estimation ideas to describe our method for differentiating between ID and OOD data based a continuous checksum and user defined threshold error.

## 3 PROPOSED METHOD: OUT-OF-DISTRIBUTION PREDICTION DETECTION WITH SOFT CHECKSUMS

Here we will consider a multi-output regression task given by a normalized and non-dimensional dataset $\mathcal{D} = \{\boldsymbol{x}_n, \boldsymbol{y}_n\}_{n=1}^N$, with $\boldsymbol{x} \in \mathbb{R}^d$ and $\boldsymbol{y} \in \mathbb{R}^k$. Standard procedure trains a neural network (NN) $f(\boldsymbol{x}; \boldsymbol{\theta})$ to predict $\hat{\boldsymbol{y}}$ by minimizing a loss function $\mathcal{L}(\boldsymbol{y}, \hat{\boldsymbol{y}})$. The dataset $\mathcal{D}$ is split into $\mathcal{D}_{\text{train}}$, used to optimize the network, and $\mathcal{D}_{\text{validation}}$, used to verify the model is not overfitting during training. $\mathcal{D}_{\text{ID}}$ includes $\mathcal{D}_{\text{train}}$, $\mathcal{D}_{\text{validation}}$, and any other data points where $f(\boldsymbol{x}; \boldsymbol{\theta})$ is a valid approximation of the true function. All other possible data points are in $\mathcal{D}_{\text{OOD}}$.

### 3.1 CHECKSUM OUTPUT

Leveraging the framework of checksums for message transmission errors, we propose a new strategy to detect model prediction errors with a single forward pass by adding one additional output node to the neural network as shown in Figure 1. Adding a check node means that the neural network predicts $(\hat{\boldsymbol{y}}, \hat{\mathbb{C}}_y)$, where $\hat{\mathbb{C}}_y$ attempts to match the checksum function $\mathbb{C}(\hat{\boldsymbol{y}})$. The user can choose the checksum function for the particular application and dataset. Given that we can always calculate a checksum error $\mathcal{L}(\hat{\mathbb{C}}_y, \mathbb{C}(\hat{\boldsymbol{y}}))$ without needing to know the true $\boldsymbol{y}$ values, we make a similar argument as energy based OOD detection (Liu et al., 2020). If the model is unable to produce a small enough checksum error, then the model is likely predicting on OOD data and should not be trusted.

In practice, the simulation workflow should answer a binary question, do we trust the prediction or not? The requirements for trusting a prediction will differ for each specific problem, as well as the cost of being wrong. In some cases, incorrectly trusting any OOD prediction (false negative) may be detrimental to the simulation, while in others there may be a level of tolerance depending on the error magnitude or number of false negatives. Similarly, some calculations may be so costly that

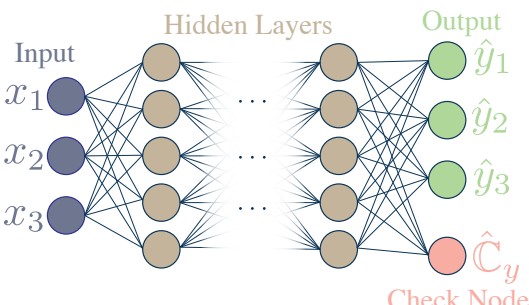

Figure 1: Adding a check node to the neural network allows the user to encode a checksum function into the output layer. We can then use the degree of violation of this function as a metric for determining prediction reliability.

not trusting an ID prediction (false positive) and requiring an unnecessary calculation is worse than limited false negatives. Every application will need to define a specific metric for the effectiveness.

We set a threshold checksum error to determine reliability, with values above this flagged to be OOD, and those below assumed reliable and ID. For the demonstration in Section 5, we require a threshold value which guarantees a 99% true negative rate on $\mathcal{D}_{\text{validation}}$, and aim for a minimum false negative rate on $\mathcal{D}_{\text{OOD}}$. We refer to the threshold checksum error as the 99% True Negative value.

Section 5 shows results using a linear checksum function (1), and sinusoid checksum function (2). While not shown here, some scientific applications may not require an additional check node as there is already a physically conserved quantity the outputs must satisfy, such as conservation of mass in a chemical kinetics surrogate model. This physical checksum could be used in place of an artificially encoded function.

$$\mathbb{C}(\boldsymbol{y}) = \sum_i y_i \tag{1}$$

$$\mathbb{C}(\boldsymbol{y}) = \sin\left(w\left|\sum_i y_i\right|\right) \tag{2}$$

### 3.2 IMPROVED LOSS FUNCTION

We aim to improve OOD detection results by explicitly incorporating the checksum function into the loss function, as shown in (3).

$$\mathcal{L} = \underbrace{\mathcal{L}_{\text{prediction}} + \mathcal{L}_{\text{checksum}}}_{\text{true - predicted mismatch}} + \underbrace{\mathcal{L}_{\text{ID}}}_{\text{ID checksum penalty}} + \underbrace{\mathcal{L}_{\text{OOD}}}_{\text{OOD checksum reward}} \tag{3}$$

$\mathcal{L}_{\text{prediction}}$ and $\mathcal{L}_{\text{checksum}}$ represent chosen loss functions to penalize inaccurate predictions of $(\hat{\boldsymbol{y}}, \hat{\mathbb{C}}_y)$ compared to the true values. Specifically, $\mathcal{L}_{\text{checksum}}$ penalizes when the check node output $\hat{\mathbb{C}}_y$ does not match the checksum function of the true output values $\mathbb{C}(\boldsymbol{y})$. $\mathcal{L}_{\text{ID}}$ slightly differs in that it penalizes when $\hat{\mathbb{C}}_y$ does not match the checksum function of the predicted values $\mathbb{C}(\hat{\boldsymbol{y}})$. Ideally, both terms would be the same, but if the predictions on the training data have error, $\mathcal{L}_{\text{ID}}$ will explicitly train the model to produce a low checksum error while $\mathcal{L}_{\text{prediction}}$ and $\mathcal{L}_{\text{checksum}}$ train the model to be accurate. This follows from similar methods used in PINNs (Raissi et al., 2019) in that we know the NN predictions should be related by the checksum function, and we directly incorporate this into the loss function. Conversely, we design $\mathcal{L}_{\text{OOD}}$ to reward violations of the checksum function on OOD data points, similar to loss terms applied in some classification problems (Hendrycks et al., 2019; Liu et al., 2020). It is important to note that often the true $\boldsymbol{y}$ value is not available for OOD

data points, and therefore $\mathcal{L}_{\text{OOD}}$ should be chosen to only depend on the predicted values with inputs from $\mathcal{D}_{\text{OOD}} = \{x'\}$.

For the experiments in Section 5, all loss terms are based on mean-squared error (MSE) with batch size $M$ as shown in (4). In order to maintain balanced terms during training, we choose $\mathcal{L}_{\text{OOD}}$ (4d) to be an inverted squared error so that the value approaches zero as the checksum error increases. We then add a small $\epsilon$ to the denominator to avoid division by zero errors in the unlikely event that the checksum error is zero.

$$\mathcal{L}_{\text{prediction}} = \frac{1}{M} \sum_{j}^{M} \left( \frac{1}{k} \sum_{i}^{k} \left( y_i^{(j)} - \hat{y}_i^{(j)} \right)^2 \right) \tag{4a}$$

$$\mathcal{L}_{\text{checksum}} = \frac{1}{M} \sum_{j}^{M} \frac{1}{k} \left( \mathbb{C}(\boldsymbol{y}^{(j)}) - \hat{\mathbb{C}}_y^{(j)} \right)^2 \tag{4b}$$

$$\mathcal{L}_{\text{ID}} = \lambda_{\text{ID}} \frac{1}{M} \sum_{j}^{M} \left( \mathbb{C}(\hat{\boldsymbol{y}}^{(j)}) - \hat{\mathbb{C}}_y^{(j)} \right)^2 \tag{4c}$$

$$\mathcal{L}_{\text{OOD}} = \lambda_{\text{OOD}} \frac{1}{\frac{1}{M} \sum_{j}^{M} \left( \mathbb{C}\left( \hat{\boldsymbol{y}}'^{(j)} \right) - \hat{\mathbb{C}}_y'^{(j)} \right)^2 + \epsilon} \tag{4d}$$

Choosing an optimal method to sample $\mathcal{D}_{\text{OOD}}$ to calculate $\mathcal{L}_{\text{OOD}}$ is not always an easy problem due to complexities in delineating ID datasets. The user may bias the model towards specific OOD data points by only including data from a limited region in the input space, or including data points in the OOD dataset that are actually ID. We avoid these issues by randomly sampling data points outside of the hypercube bounding $\mathcal{D}_{\text{training}}$. These data points are not sampled from simulations or experiments and are well outside the maximum possible bounds of the training data by construction. This procedure seeks to suppress biases or invalid inclusions when calculating $\mathcal{L}_{\text{OOD}}$.

## 4 NUMERICAL EXPERIMENT

We demonstrate the effectiveness of checksum errors as an OOD detector for a surrogate model of Non-Local Thermodynamic Equilibrium (NLTE) calculations, a key step in atomic kinetics and radiation transport calculations. Applications span a variety of fields, including inertial confinement fusion (ICF), magnetic fusion, X-ray lasers and laser-produced plasmas. With respect to ICF specifically, the atomic physics code *Cretin* (Scott, 2001) carries out the NLTE calculations, taking ten to ninety percent of the total simulations wall clock time (Kluth et al., 2020).

The machine learning surrogate model takes in the electron density, temperature, and radiation spectrum, and predicts the absorption spectrum. For our example, each spectrum is each defined by 85 frequency bins, resulting in an 87 dimensional input space and 85 dimensional output space. We generate the dataset by running many ICF simulations in *Cretin*, and manually divide ID data for training and validation, and OOD data for evaluating the soft checksum metric, as shown in Figure 2. In this way, we are only considering if a soft checksum can flag the more difficult and relevant subset of $\mathcal{D}_{\text{OOD}}$ that is realistic for the surrogate model to encounter in a simulation.

For this study, we conducted a limited parameter sweep to determine the optimal hyperparameters for the given experiment. Importantly, this was not a general method of selecting the hyperparameters and depended on the chosen OOD dataset. Specifically, we set $\lambda_{\text{ID}}$ and $\lambda_{\text{OOD}}$ to $0.01$. To calculate $\mathcal{L}_{\text{OOD}}$, we sample a subset of $\mathcal{D}_{\text{OOD}}$ with values between $20\%$ to $25\%$ outside of the hypercube bounding $\mathcal{D}_{\text{training}}$. When encoding (2) as the checksum function, we set $w = 0.0001$ to achieve a nonlinear relationship while also maintaining a low enough frequency that $\mathbb{C}(\hat{\boldsymbol{y}})$ is not effectively random noise.

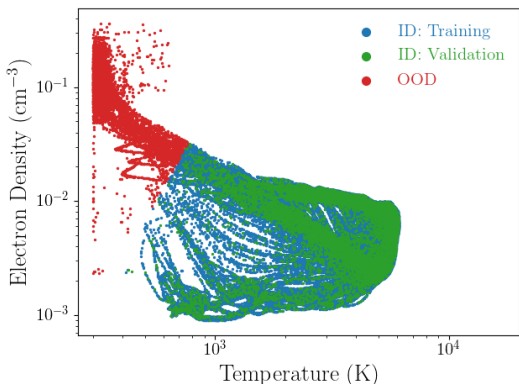

Figure 2: We generated the training, validation and out-of-distribution (OOD) datasets from trusted *Cretin* simulations (Scott, 2001). While the data has 87 dimensions, we split the OOD data points with an arbitrary dividing line in the density-temperature plane to create a set of data points not shown to the surrogate model in training by construction.

Table 1: False Negative at 99% True Negative Rates (FNR99) for specific loss functions and checksum functions. Lower is better. In each case we show results from a neural network optimized through a limited parameter sweep.

| Loss Function | FNR99 (%) | |
|---|---|---|
| | $\mathbb{C}(\boldsymbol{y}) = \sum y_i$ | $\mathbb{C}(\boldsymbol{y}) = \sin\left(w\left|\sum y_i\right|\right)$ |
| $\mathcal{L}_{\text{prediction}} + \mathcal{L}_{\text{checksum}}$ | 8.93 | 3.84 |
| $\mathcal{L}_{\text{prediction}} + \mathcal{L}_{\text{checksum}} + \mathcal{L}_{\text{ID}}$ | 11.08 | 6.31 |
| $\mathcal{L}_{\text{prediction}} + \mathcal{L}_{\text{checksum}} + \mathcal{L}_{\text{OOD}}$ | **4.76** | **1.64** |
| $\mathcal{L}_{\text{prediction}} + \mathcal{L}_{\text{checksum}} + \mathcal{L}_{\text{ID}} + \mathcal{L}_{\text{OOD}}$ | 13.64 | 7.30 |

## 5  DISCUSSION

As introduced in Section 3.1, we measure OOD detection effectiveness using a False Negative at 99% True Negative rate (FNR99). This represents the percentage of OOD predictions that have a checksum error less than the 99% True Negative value and our method would incorrectly not flag.

Table 1 reports FNR99 rates for soft checksums implemented with four different loss functions. As shown, simply adding the check node to encode a checksum function without including $\mathcal{L}_{\text{ID}}$ or $\mathcal{L}_{\text{OOD}}$ achieves strong separation between ID and OOD data points. We can further improve the separation by including $\mathcal{L}_{\text{OOD}}$ in the loss function, effectively training the model to increase checksum errors on OOD predictions. As shown in Figure 3, in addition to separation between ID and OOD data, there is also a linear correlation between checksum error and prediction error for OOD data. The correlation between errors potentially allows for soft checksums to not only flag OOD predictions, but also serve as a proxy for prediction error.

On the other hand, including $\mathcal{L}_{\text{ID}}$ in the loss function surprisingly has a negative effect on the separation. When including both $\mathcal{L}_{\text{ID}}$ and $\mathcal{L}_{\text{checksum}}$ and assuming there is non-zero error, we concurrently train the surrogate to predict two different values of $\hat{\mathbb{C}}_y$. $\mathcal{L}_{\text{checksum}}$ pushes the prediction towards $\mathbb{C}(\boldsymbol{y})$, while $\mathcal{L}_{\text{ID}}$ pushes the prediction towards $\mathbb{C}(\hat{\boldsymbol{y}})$. This could explain the decrease in performance and presents an opportunity to better understand how the surrogate model learns the checksum function and the conflict between the two terms.

Our proposed method of applying soft checksums to flag untrustworthy predictions shows promising results and deserves more in-depth study. While considerably cheaper and simpler to implement than many current state-of-the-art OOD detection methods, we must also conduct benchmark comparisons to establish the relative effectiveness. As part of these comparisons, there are areas for improvement to investigate.

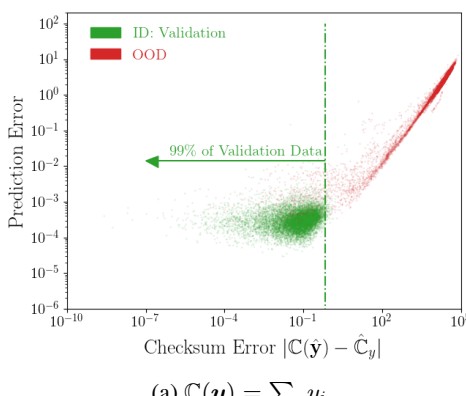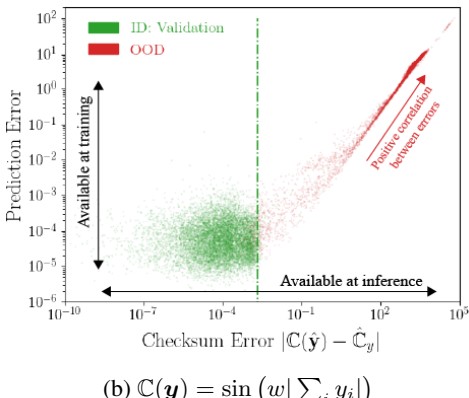

(a) $\mathbb{C}(\boldsymbol{y}) = \sum_i y_i$            (b) $\mathbb{C}(\boldsymbol{y}) = \sin\left(w|\sum_i y_i|\right)$

Figure 3: Relationship between checksum error and prediction error with an optimized loss function, and either a summation (3a) or sinusoid (3b) checksum function. We determine reliability based on a threshold checksum error with 99% of the validation data below this value. With respect to out-of-distribution data points, we see a positively correlated relationship between the checksum and prediction errors.

1. The checksum function is currently not optimized. An ideal checksum function is complex enough that the ML surrogate model cannot memorize it for any given inputs, but is not too complex that it cannot learn the function for ID inputs. For the sinusoid checksum function (2), varying the frequency hyperparameter $w$ spans both of these conditions. If it is set too high, then the ML model sees effectively random noise, but if set too low, it reduces to a simpler linear or constant function.

2. Adding multiple check nodes to encode multiple checksum functions should better reveal the edges of the domain of validity for a given surrogate model. While it is possible that the model learns to memorize one checksum function and produce a low checksum error on OOD data, it is less likely that this is the case for multiple checksum functions. Adding redundancies will reduce the possibility of coincidentally low checksum errors.

3. Incorporating OOD data points in the training process improves OOD detection, but has limitations. Optimizing the distance outside of the hypercube to sample $\mathcal{D}_{\text{OOD}}$ requires a balance between being close enough to the boundary to improve ID and OOD separation, but far enough away to avoid difficulty training the surrogate model. Additionally, while sampling outside of a bounding hypercube guarantees there is no overlap with $\mathcal{D}_{\text{training}}$, it also misses potential OOD regions within the hypercube and holes within the training dataset. There is an opportunity to better capture these regions and improve OOD detection.

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
