# OpenReview forum: "Soft Checksums to Flag Untrustworthy Machine Learning Surrogate Predictions and Application to Atomic Physics Simulations"
_ICLR.cc/2025/Conference — ICLR 2025 Conference Withdrawn Submission_

### Official Review · Reviewer_ZiUe · 2024-11-03

**Soundness:** 2
**Presentation:** 3
**Contribution:** 2
**Rating:** 3
**Confidence:** 4

**Summary:**

The authors present a soft-checksum method to detect whether the input to the neural network is in-distribution or out-of-distribution of the training dataset. The checksum can be computed simply just by summing the output signal, or sin of absolute sum of the signal. They add a final layer to the neural network to predict what the checksum would be (without any information of the true output value). During the training, they optionally added several terms involving the checksum:
1. to make the predicted checksum **as close** as possible to the *computed checksum of the true output* for in-distribution data,
2. to make the predicted checksum **as close** as possible to the *computed checksum of the predicted output* for in-distribution data, and
3. to make the predicted checksum **as far** as possible to the *computed checksum of the predicted output* for out-of-distribution data.
The authors did an empirical experiment using plasma physics simulation software, Cretin, and show that the checksum can be used to detect out-of-distribution data.

**Strengths:**

1. Computing checksum and predicting checksum can be done relatively easily and computationally cheap.
2. The results presented show positive results on the tested case.

**Weaknesses:**

1. Lack of diversity of the test case and architecture. The presented Numerical Experiment section only discusses one test case using Cretin. The details of the NN architecture is also unclear. The lack of diversity in test case and NN architecture makes it unclear whether the method will work with other test cases and NN architectures. Preferably some test cases with relatively high complexity with relatively large NN (maybe ~100k parameters) would improve the strength of the paper.
2. Non-robust test done on OOD data. In the paper, the authors mention that OOD distribution is done by simulating parameters that are 20-25% outside the bounding box of the parameters they previously set. Including the OOD distribution in the training distribution makes it somehow to be *"in-distribution"*. This raises concern whether the true OOD (OOD data that is not in the training data or generated by significantly different way) can still be detected. For example, histograms that are randomly generated (not from the simulations with OOD parameters), histograms that has extreme values, etc.
3. Lack of comparison to other OOD detection method. The experimental section did not compare the presented method with other OOD detection methods.

**Questions:**

1. Why does the inclusion of $\mathcal{L}_{ID}$ seems to make it worse?
2. The details of the architecture and training procedure used in the experiment seems missing. What is the architecture used for the experiment and the training procedure for the experiment?
3. The checksum seems to only consist of a single number. Would make it a vector (instead of a single number) improve the OOD detection?

---

### Official Review · Reviewer_Mtmw · 2024-11-04

**Soundness:** 1
**Presentation:** 2
**Contribution:** 2
**Rating:** 3
**Confidence:** 4

**Summary:**

This paper proposes soft checksums to flag untrustworthy predictions in the context of physics simulations. The authors propose to add a checksum node to the model as an additional optimization objective during training. The experimental results with Cretin simulations show low FNR99 (False Negative at 99% True Negative Rates), supporting the claim of the paper.

**Strengths:**

The paper is well-presented and easy to follow.

The idea of adapting checksums from data transmission to out-of-distribution (OOD) detection in deep learning is novel.

The soft checksum method is computationally efficient and introduces little overhead for training and inferencing.

**Weaknesses:**

The motivation of the checksum function choice is unclear.

The paper lacks comparisons in existing works on OOD detection / uncertainty estimation. For example, the classical Bayesian methods mentioned in section $2.1$ (MC Dropout, Deep ensemble, etc.) can be compared.

The experiment section is not very persuasive, not only for the lack of comparison of methods, but also for the lack of tasks. There is only 1 task included in the experiment section, while many other tasks are available for OOD detection setups. For example, the PDE simulations (Heat, Poisson, etc.) are good fits since the boundary conditions and initial conditions can be partially isolated to construct OOD setup.

**Questions:**

The term $k$ in equation $4b$ is not defined, is it the same as in $4a$, representing the batchsize?

For equation $4b$ and $4c$, the only difference is replacing the batch averaging with a predetermined weight. What is the weight used on the experiment? Is it similar with batch averaging term?

For other questions, please refer to the weakness part.

---

### Official Review · Reviewer_12w9 · 2024-11-05

**Soundness:** 1
**Presentation:** 2
**Contribution:** 1
**Rating:** 1
**Confidence:** 5

**Summary:**

The motivation behind the work proposed in this paper is common among most out-of-distribution (OOD) data detection methods in machine learning: deploying an ML model could lead it to encounter near-in-distribution (NID) or OOD data. Evidently, this can cause severe errors in its ability to fulfil its intended task correctly. The authors propose a method, inspired by checksum functions, to add another output node to the neural network and modify the loss function to encourage the checksum node to flag OOD data.

**Strengths:**

- Even though the proposed work lacks significant novelty, I do believe that there is some promise behind it. The, albeit limited results, show signs of the method working reasonably and with some further/major refinement could offer a nice solution.
- Figures throughout are useful and relevant. They are a positive inclusion.
- The discussion section is transparent and honest, openly addressing both the strengths and limitations of the study. This level of frankness is commendable.

**Weaknesses:**

- Firstly, the work seems to be built upon work in [1]. The paper includes no reference to the work, and the novelty past it is very limited.
- The 'Related work & Background' section is severely limited. In the uncertainty quantification discussion, there are no remarks to very popular methods such as conformal prediction, test-time augmentation, evidential deep learning, variational inference, etc. I find this poor especially since some parallels can be drawn between EDL and this paper.
- The writing in the paper would benefit from improved organization and clarity. Currently, certain arguments or points appear prematurely in earlier sections (e.g., points relevant to Section 5 are raised in Section 3), which disrupts the logical flow of the content. This causes the text to read unevenly and creates an informal tone that may detract from the paper’s scientific rigour. I recommend restructuring to ensure each section clearly builds on the previous one.
- The use of the sum of predictions as a checksum for identifying in-distribution (ID) or out-of-distribution (OOD) data lacks sufficient mathematical rigour and robustness. While the authors imply that this approach checks for consistency to distinguish between ID and OOD cases, the method appears overly simplistic and insufficiently substantiated. Without clear justification or evidence supporting the effectiveness of this metric, it is unclear how well it can reliably capture the complexity of OOD detection.
- While the application of the proposed approach to Non-Local Thermodynamic Equilibrium (NLTE) calculations is both valid and relevant, it would benefit from broader experimental validation. Given the generalizable nature of the method, an additional evaluation on standard machine learning datasets or synthetic data would strengthen the study's claims and allow for a more comprehensive assessment of its applicability across domains. Benchmarking on common or synthetic datasets would also provide clearer comparisons to existing methods and better illustrate the versatility and robustness of the approach.
- In Figure 2, the boundary between ID and OOD data is abrupt, despite minimal differences near the threshold. Introducing a near-ID (NID) dataset could create a gradient at the boundary, allowing for a more nuanced transition and improving the model’s handling of data close to this threshold. I expect that the majority of false negatives reported in Table 1, could potentially show as NID data instead, therefore improving your results and increasing transparency into the limitations of the work.
- Was any ablation conducted on the threshold value to assess its impact? Were results better at a slightly lower threshold?
- The paper claims a novel method for exposing the model to random OOD data without biasing it toward a limited region, but there is no clear evidence or demonstration of this in the results. This claim would benefit from supporting experiments or explanations to substantiate it.

[1] Możejko, M., Susik, M. and Karczewski, R., 2018. Inhibited softmax for uncertainty estimation in neural networks. arXiv preprint arXiv:1810.01861.

Small weaknesses
- Separating the conclusion of the paper from the Discussion section could be useful for the readability of the paper.
- The paper devotes considerable attention to explaining basic, well-understood concepts, such as the use of a validation dataset. This level of detail seems unusual, as it may be redundant for the intended audience.

**Questions:**

Can the authors comment on the following:
- Can you clarify how the sum of predictions as a checksum value reliably distinguishes between in-distribution (ID) and out-of-distribution (OOD) data? Is there a theoretical basis or empirical evidence to support its robustness?
- Have you considered validating the approach on standard machine learning or synthetic datasets to demonstrate its broader applicability?
- The paper mentions a novel method for exposing the model to random OOD data without biasing it. Could you provide more details or experimental evidence for this claim?
- Was any ablation conducted on the threshold values used in the model?
- Could you explore creating a "near ID" dataset to provide a smoother transition at the ID-OOD boundary and possibly improve results?

---

### Note · Authors · 2024-11-20

I have read and agree with the venue's withdrawal policy on behalf of myself and my co-authors.